# Reduction in diarrhea cases following implementation of COVID-19 hand hygiene interventions in Ghana: A causal impact analysis

**George Asumah Adu** [1]*, **Kingsley E. Amegah**[2], **Henry Ofosu Addo**[3], **Theresa Andoh**[4], **Ferguson Duvor**[1], **Gladys Antwi**[5], **Nana Yaw Peprah**[1], **Ernest Kenu**[5], **Franklin Asiedu Bekoe**[1], **Keziah L. Malm**[1]

**1** Public Health Division, Ghana Health Service, Accra, Ghana, **2** Department of Data Science and Economic Policy, University of Cape Coast, Cape Coast, Ghana, **3** Department of Industrial and Health Sciences, Takoradi Technical University, Sekondi-Takoradi, Ghana, **4** Department of Nutrition and Food Science, University of Ghana, Accra, Ghana, **5** Epidemiology Department, GFELTP, School of Public Health, University of Ghana, Accra, Ghana

* georgeaduasumah@yahoo.com

**Data Availability Statement:** The data underlying the results presented in the study have been attached as supplementary information.

## Abstract

### Background

The human hand has constant contact with the environment, hence requires regular hand hygiene. Hand hygiene has gained recognition because of the COVID-19 pandemic and is a largely effective, affordable preventive measure against infectious diseases. This study used both national and sub-national analyses to evaluate the effect of COVID-19 handwashing guidelines on instances of diarrhea in Ghana.

### Methods

Data on diarrhea cases spanning February 2018 and March 2022 were retrieved from the District Health Information Management System (DHIMS 2) using a data extraction guide. The data were summarized using descriptive statistics. The difference in diarrhea cases between the pre-COVID-19 and COVID-19 periods was measured using a two-sample t-test across Ghana's 16 administrative areas. Causal Impact package in R statistical software was employed to determine the impact of the introduction of COVID-19 hand hygiene protocols on diarrheal disease.

### Results

A total of 5,645,533 diarrheal cases reported between February 2018 and March 2022 through the routine MIS (DHIMS2) were examined. Fifty-three percent of the cases occurred before the introduction of the hand hygiene protocol. Descriptive statistics indicated a statistically significant decrease in average diarrheal cases during the hand hygiene implementation era (13,463 cases reduction, p<0.001). Sub-national analyses revealed significant reductions in various regions: Greater Accra, Ashanti, Ahafo, Central, Eastern, Northern,

**Funding:** The author(s) received no specific funding for this work.

**Competing interests:** The authors have declared that no competing interests exist.

Upper East, Upper West, and Volta (p<0.05). Causal impact analysis confirmed 11.0% nationwide reduction in diarrheal cases attributed to the COVID-19 hand hygiene protocols (p<0.001).

## Conclusion

This study underscores the effectiveness of COVID-19 hand hygiene protocols in reducing diarrheal morbidity in Ghana, with varying regional impacts. These findings advocate for the sustenance of investments and commitments made at the COVID hand hygiene protocols, particularly in this era where the pandemic appears controlled.

## Introduction

Public health interventions in times of global health crises have been noted to be very impactful on health systems [1,2]. The global health infrastructure has undergone substantial modifications since the introduction of COVID-19. The COVID-19 pandemic, which was initially discovered in Wuhan, China, in late 2019, affected nearly every region of the planet, taking many lives and having tremendous impact on all facets of human society [3]. As of 8th June 2023, more than 690, 032, 285 cases of COVID-19 have been recorded with 6,889, 153 deaths globally [4]. Ghana reported the first two cases of COVID-19 on March 12th, 2020 [5]), and as of April 2023, a total of 171, 657 cases of COVID-19 with associated 1,462 deaths had been recorded in Ghana [6]. To assuage the escalating cases and death toll from COVID-19, the Government of Ghana established several measures based on the WHO's recommendations, involving physical distancing, regular handwashing with soap under running water and rubbing of hands with alcohol-based sanitizers [5,7].

The COVID-19 pandemic has shone a light on hand hygiene as an inexpensive, widely applicable response [3]. Hand hygiene remains a significant component in preventing many transmissible infections including COVID-19 and diarrheal diseases [8–11]. It generally refers to any action involving hand cleansing with the principal objective of removing dirt, organic material and/or microorganisms [12,13]. Numerous microorganisms have been found to have their richest habitat in the human palm including those associated with faeco-oral diseases, highlighting the importance of hand cleanliness in infection prevention [14,15]. In underdeveloped nations, diarrheal illnesses continue to be a serious public health concern. It remains one of the commonest preventable causes of morbidity and mortality, responsible for 1.6 and over 0.5 million deaths among all age groups and children under 5 years, respectively [16]. In Ghana, like other low-income countries, diarrhea ranks among the top ten causes of morbidity and mortality, killing more than 14,000 children annually in Ghana [15,17]. Jin et al. (2020) defined diarrhea as the passing of loose stools more than three times per day. Diarrhea is commonly caused by bacteria such as *Escherichia coli* and *Vibrio cholerae* [18,19]; viruses such as rotavirus and adenovirus and parasites such as *Giardia* [20].

Maintaining good hand hygiene is still one of the best strategies to get rid of pathogens and stop the spread of infectious diseases. To stop the spread of COVID-19, it was stressed how important it was to maintain personal cleanliness, especially when it came to washing your hands with soap or alcohol-based sanitizer [21,22]. Though COVID-19 and many diarrheal diseases share different aetiologic agents, they have one thing in common: using the intervention of hand hygiene to break the cycle of infection. The habit of hand hygiene has recorded a significant improvement around the world since the onset of COVID-19 [11].

In Ghana, hand hygiene was declared mandatory and a pre-requisite for entering public places as part of the COVID mitigating strategies [5]. Several studies have documented the benefits of hand hygiene on diarrheal diseases [23,24]. But till date, there is a paucity of research looking at the effects of this hand hygiene practice instituted during the COVID-19 pandemic on diarrheal cases. This study, therefore, fills that lacuna by drawing on diarrheal disease data before and during the COVID-19 pandemic to assess the effects of COVID hand hygiene protocols on diarrheal diseases in Ghana.

## Methods

### Study design

This was a descriptive cross-sectional study that examined diarrhea cases between February 2018 and March 2022 using secondary data obtained from Ghana's District Health Information Management System (DHIMS 2). DHIMS 2 is an integrated web-based electronic database that is used for the storage and management of aggregated data across health facilities in Ghana.

### Study site

Ghana is a West African country sharing border with Cote d'Ivoire to the west, Burkina Faso to the north, Togo to the east, and the Gulf of Guinea and the Atlantic Ocean to the south (Fig 1). Ghana consists of a total of sixteen administrative regions and 261 districts. The Greater Accra (17.7%), Ashanti (17.6%), Eastern (9.5%), and Central (9.3%) regions comprise almost half of Ghana's population, which is estimated to be 31 million [25]. Ghana's average household size is 3.6 and its annual population growth rate is 2.1% [25]. Most health institutions in Ghana are publicly owned, although they receive help from private, faith-based, traditional, and alternative service providers which help the healthcare system in Ghana. Health services are delivered at diverse levels: tertiary-level (teaching hospitals), secondary-level (regional and district hospitals), and primary- (health centers, and Community-based Health Planning Services (CHPS). Almost all levels of facilities manage diarrhea cases and enter aggregated data into DHIMS 2 database monthly.

### Data source and variables

In Ghana, all healthcare facilities are required by the Ministry of Health (MoH) and the Ghana Health Service (GHS) to capture, and report aggregated data on all services provided on specific reporting forms every month. These data are validated by a data validation team in each healthcare facility before the 5th of the ensuing month for accuracy and validity before feeding into the District Health Information Management System (DHIMS 2) database. Healthcare facilities in Ghana during the COVID-19 pandemic continuously collated, validated, and reported data on essential healthcare indicators including cases of diarrheal disease into DHIMS 2. On the 19th of June 2023, monthly data on diarrhea cases were extracted from DHIMS 2 spanning the period from February 2018 to March 2022 for all 16 regions of Ghana to characterize the disease by place and time.

### Statistical analysis

Data on diarrheal disease were extracted from DHIMS 2 and cleaned using Microsoft Excel 2016 and later imported in R software for analysis. To be able to measure the effect of hand hygiene protocol on diarrheal disease, the data were divided into the pre-COVID period (February 2018 to February 2020) and the COVID period (March 2020 to March 2022). Data were

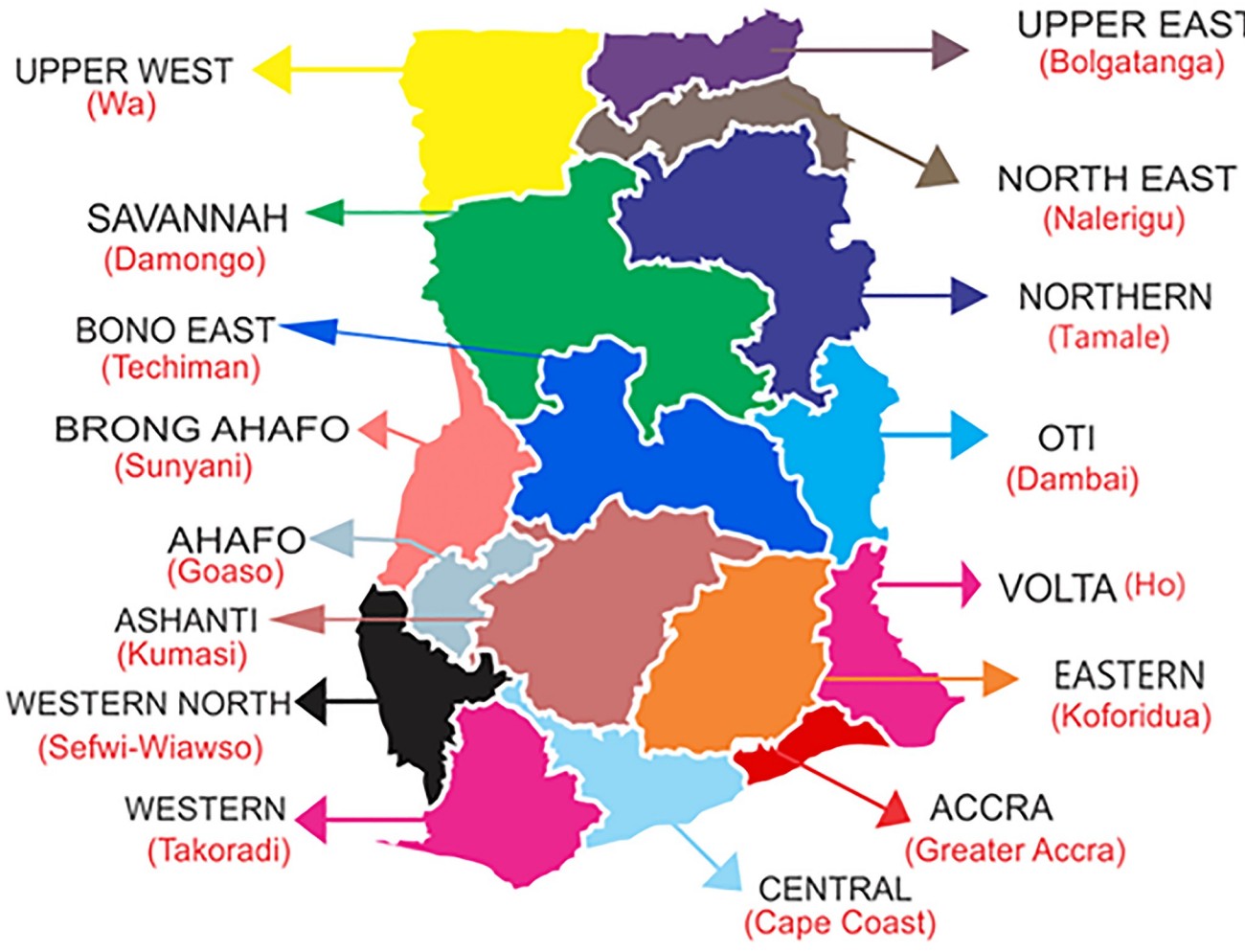

**Fig 1. Map of Ghana [26].**

summarized using descriptive statistics such as mean with standard deviation and median with interquartile range. Levene's test for equality of variances and Shapiro Wilks test for normality of the number of diarrhea cases between the pre-and-post COVID-19 hand hygiene period were conducted, after which a two-sample t-test and Wilcoxon rank-sum test were conducted to estimate the effect of COVID-19 hand hygiene protocols on diarrhea cases across the 16 regions in Ghana. Further, Google's causal impact package in R [27] which is designed for interrupted time series (ITS), was used to evaluate the effect of an intervention or event by comparing observed outcomes to a counterfactual scenario (what would have happened if the COVID-19 hand hygiene protocol had not been implemented) was used to estimate the causal effect of the introduction of the COVID-19 hand hygiene protocol on diarrhea cases. The Causal Impact package utilized a Bayesian Structure Time Series (BSTS) model to estimate the

counterfactual scenario, which represents what would have happened to diarrhea cases without the introduction of the hand hygiene protocol. This model considered both the pre-intervention (February 2018 to February 2020) and intervention (March 2020 to March 2022) periods and incorporated covariates information. The Outpatient department (OPD) attendance was identified as a potential confounding covariate and controlled for because it reflects health-seeking behavior which can influence the number of diarrheal cases recorded. Including the out-patient department attendance as a covariate was to rule out/mitigate OPD attendance as a possible cause of any observed differences. All statistical analyses were considered statistically significant at a 95% Confidence Interval with a 5% level ($p < 0.05$) of significance.

### Ethical considerations

This study utilized only aggregated secondary data and was carried out as part of efforts to improve quality of both COVID-19 and diarrheal diseases management in Ghana.

This study aligns with Helsinki's ethical guidelines on the use of secondary data and did not require ethical approval since the study only used aggregated secondary data which did not contain personal identifiers of patients [28].

This research did not involve any direct interaction or involvement with human participants; instead, the data used in the study were taken from pre-existing datasets and repositories that had no personal identifiers. It is also important to note that the secondary data used in this study came from reliable sources that follow high ethical standards and procedures for data gathering and distribution [29].

The researchers are devoted to advancing accountability and transparency in research techniques, as well as to making knowledge contributions that advance society while upholding the participants' rights to privacy.

## Results

### Descriptive statistics and differences in diarrhea cases before and during the introduction of COVID-19 hand hygiene protocol in Ghana

A total of 5,645,533 diarrhea cases were reported in Ghana from February 2018 to March 2022. Before the introduction of the COVID-19 hand hygiene protocol, a total of 2,991,054 diarrhea cases were reported, accounting for about 53% of all diarrhea cases reported for the period under study. The median diarrhea cases reported before the introduction of the protocol was 121,383 with an inter-quartile range (IQR) of 13,869 cases with a minimum and maximum of 97,557 and 139,162 cases, respectively. During the observation of the COVID-19 hand hygiene protocol, the median reported diarrheal cases was 106,155 with an IQR of 12,744 with 78,861 as the minimum diarrhea cases reported and 148,273 as the maximum cases reported for the period (**Table 1**). A two-sample t-test confirmed a significant difference between the average number of diarrhea cases reported before and during the introduction of the COVID-19 hand hygiene protocol in Ghana (difference = 13,463, 95% CI [5847.19, 21,078.81], t (48) = 3.55, p = 0.001; Cohen's d = 1.01, 95% CI [0.41, 1.59]) (**Fig 2**).

**Table 1. Descriptive statistics of diarrhea cases in Ghana.**

| Period | Sum | Mean | SE | SD | Median | IQR | Min. | Max. |
|---|---|---|---|---|---|---|---|---|
| Pre-protocol | 2,991,054 | 119,642 | 2,257 | 11,336 | 121383 | 13869 | 97,557 | 139,162 |
| Protocol | 2,654,479 | 106,179 | 3,034 | 15,171 | 106155 | 12744 | 78,861 | 148,273 |
| Ghana | 5,645,533 | 112,910 | 2,106 | 14,896 | 111088 | 20104 | 78,861 | 148,273 |

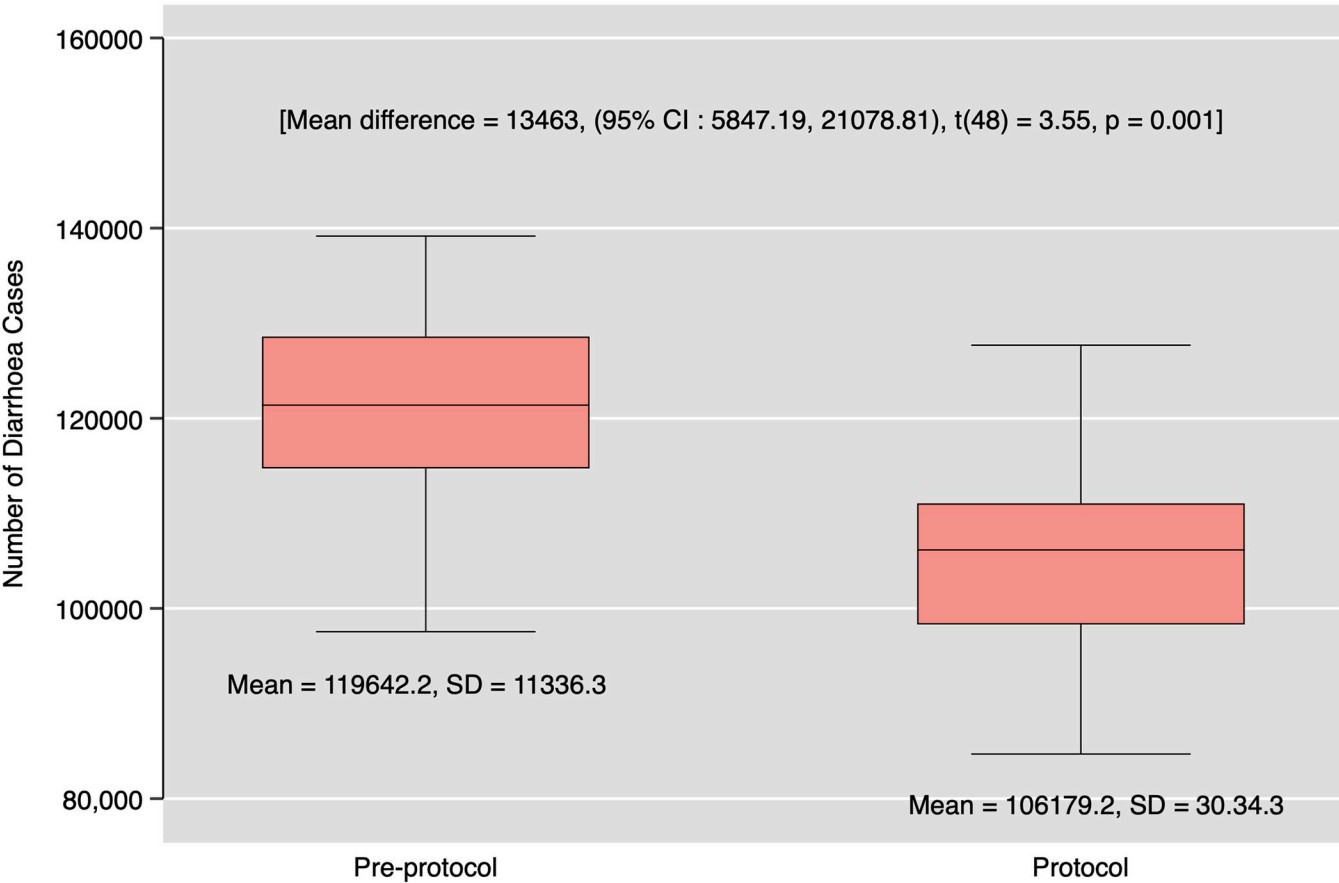

**Fig 2. Difference in diarrhea cases before and during the introduction of the COVID-19 handwashing protocol in Ghana.**

### Sub-national analysis of differences in diarrhea cases before and during the introduction Of COVID-19 handwashing protocol in Ghana

Table 2 presents the effect of COVID-19 hand hygiene protocol on diarrhea cases across the 16 regions in Ghana. In the Ahafo region, there was a statistically significant reduction in diarrhea cases (Mean difference = 429.3, CI: 155.9–702.8, p = 0.003) with a Cohen's d = 0.89, indicating a large effect. Statistically significant reductions in diarrhea cases were observed in the Central, Eastern, Greater Accra, Northern, Upper East, Upper West, Volta, and Western regions, with varying effect sizes, particularly, the Volta region with the highest effect size (Cohen's d = 1.31). Marginal reductions in diarrhea cases were observed in the Ashanti and Bono regions, however, these reductions were not statistically significant (p = 0.074 and p = 0.097, respectively). The North East region observed a non-significant increase in diarrhea cases (Mean difference = -493.3, p = 0.109), as also, observed in Oti and Savannah regions. Country-wide, there was a statistically significant reduction (Mean difference = 841.4, CI: 344.2–1338.6, p = 0.001, Cohen's d = 0.23), indicating that the introduction of hand hygiene protocol had a moderate effect on the reduction of diarrhea cases in Ghana.

### Impact of COVID-19 hand hygiene protocol on diarrheal disease in Ghana

The impact of the COVID-19 hand hygiene protocol on diarrhea case incidence varied across regions in Ghana. Nationwide, the average number of diarrhea cases after the introduction of

**Table 2. Mean differences in cases of diarrheal disease pre-and post COVID-19 hand hygiene protocol in Ghana.**

| Regions | Pre-Protocol Period | | | Post-Protocol Period | | | Average Effect | | | | Effect Size |
|---|---|---|---|---|---|---|---|---|---|---|---|
| | Mean | Std.Dev | 95% CI | Mean | Std.Dev | 95% CI | Difference | 95% CI | t/z | P-value | Cohen's d |
| Ahafo | 4256.6 | 435.7 | 4076.8–4436.45 | 3827.3 | 522.1 | 3611.8–4042.8 | 429.3 | 155.9–702.8 | 3.16 | **0.003** | 0.89 |
| Ashanti | 15327.9 | 4994.9 | 13266.1–17389.7 | 13357.5 | 2456.2 | 12343.6–14371.3 | 1970.4 | -267.9–4208.7 | 1.80[z] | 0.074 | 0.50 |
| Bono | 7803.5 | 1046.9 | 7371.4–8235.7 | 7275.9 | 1155.4 | 6799.0–7752.8 | 527.6 | -99.4–1154.6 | 1.69 | 0.097 | 0.48 |
| Bono East | 7430.4 | 1195.7 | 6936.9–7924.0 | 7444.0 | 1156.0 | 6966.8–7921.2 | -13.6 | -682.4–655.2 | -0.04 | 0.968 | -0.01 |
| Central | 8983.5 | 1022.8 | 8561.3–9405.7 | 7759.1 | 1062.0 | 7320.7–8197.5 | 1224.4 | 631.5–1817.3 | 4.15 | **0.001** | 1.17 |
| Eastern | 13772.0 | 1753.3 | 13048.2–14495.7 | 12519.8 | 2152.8 | 11631.2–13408.4 | 1252.2 | 135.6–2368.6 | 2.26 | **0.029** | 0.64 |
| Greater Accra | 8916.0 | 1467.20 | 8310.3–9521.6 | 7169.9 | 1499.9 | 6550.8–7789.1 | 1746.0 | 902.3–2589.8 | 3.74[z] | **0.001** | 1.18 |
| North East* | 3554.7 | 701.2 | 3265.3–3844.2 | 4048.0 | 1329.0 | 3499.4–4596.6 | -493.3 | -1102.6–116.0 | -1.64 | 0.109 | -0.46 |
| Northern | 9126.9 | 1332.0 | 8577.1–9676.7 | 7426.1 | 1698.5 | 6725.1–8127.3 | 1700.7 | 832.7–2568.7 | 3.37[z] | **0.001** | 1.11 |
| Oti | 3396.2 | 559.0 | 3165.4–3626.9 | 3233.2 | 434.2 | 3054.0–3412.5 | 162.9 | -121.7–447.5 | 1.15 | 0.256 | 0.33 |
| Savannah* | 2688.3 | 372.4 | 2534.6–2842.0 | 2549.9 | 609.9 | 2298.1–2801.7 | 138.4 | -150.6–427.3 | 0.97 | 0.339 | 0.27 |
| Upper East | 6938.6 | 1307.0 | 6399.1–7478.1 | 5983.7 | 1133.2 | 5516.0–6451.5 | 954.9 | 259.3–1650.5 | 2.76 | **0.008** | 0.78 |
| Upper West | 5528.1 | 1000.3 | 5115.2–5941.0 | 4876.5 | 935.9 | 4490.2–5262.8 | 651.6 | 100.8–1202.5 | 2.38 | **0.021** | 0.67 |
| Volta | 5932.5 | 891.1 | 5564.6–6300.3 | 4726.6 | 950.8 | 4334.1–5119.0 | 1205.9 | 681.9–1729.9 | 4.63 | **0.000** | 1.31 |
| Western | 10196.2 | 810.1 | 9861.8–10530.6 | 9270.6 | 1279.1 | 8742.6–9798.6 | 925.6 | 316.7–1534.4 | 3.06 | **0.004** | 0.86 |
| Western North | 5790.8 | 2503 | 4757.6–6824.0 | 4711.0 | 2126.5 | 3833.2–5588.7 | 1079.8 | -240.9–2400.6 | 1.60[z] | 0.111 | 0.46 |
| **Ghana*** | 7477.6 | 3842.8 | 7099.9–7855.4 | 6636.2 | 3300.6 | 6311.8–6960.6 | 841.4 | 344.2–1338.6 | 3.30[z] | **0.001** | 0.23 |

\* Two-sample t-test with unequal variances

**[z] Z statistics from Two-sample Wilcoxon rank-sum (Mann–Whitney) test.**

the COVID-19 hand hygiene protocol was 106179. By contrast, the absence of the protocol would have resulted in an average of 119283 (95% CI: 114931, 123638) cases. The Absolute effect of the introduction of the protocol was a reduction in diarrhea cases by -13104 (95% CI: -17459, -8752). The relative effect expresses this reduction as a percentage, indicating a -11.0% (95% CI: -14%, -7.8%) decrease in diarrhea cases in Ghana (**Fig 3**). Resultant test statistics revealed a high statistical significance (p<0.001) and a probability of causal effect of 99.9%. In

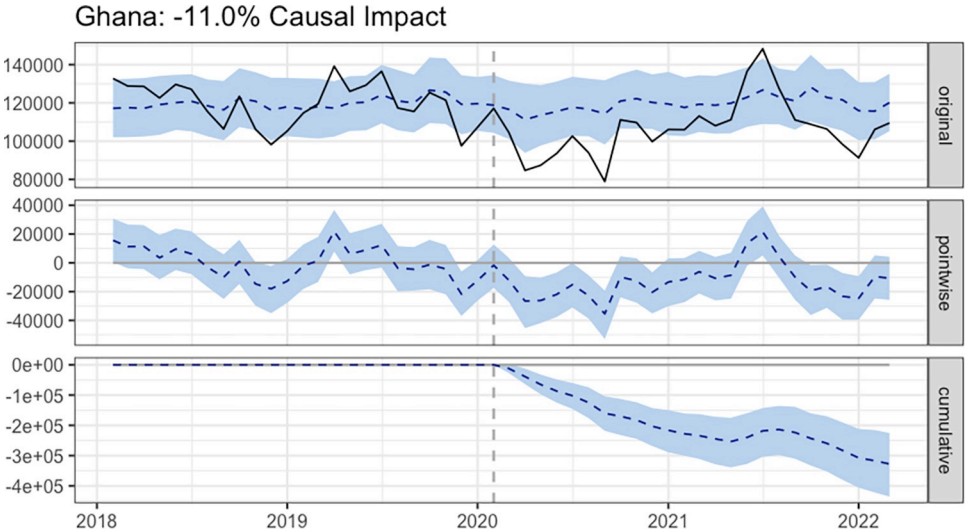

**Fig 3. Impact of COVID-19 hand hygiene protocol on diarrhea cases in Ghana.**

the Ahafo region, a significant reduction of 6.5% (p-value: 0.001) in diarrhea cases was observed, demonstrating the protocol's effectiveness. Similarly, in the Central and Eastern regions, the protocol had significant positive impacts, with reductions of 10% and 11%, respectively (both p-values: 0.001). The Greater Accra region experienced the most substantial impact, with a significant 24% reduction (p-value: 0.001). The Northern region followed with a 19% reduction (p-value: 0.001), also experienced a significant reduction. Ashanti, Upper East, and Upper West regions demonstrated reductions of 11%, 11%, and 10%, respectively (p-values: 0.003), while the Volta region saw a 14% reduction (p-value: 0.001) (**Table 3**).

## Discussion

Ghana, like many countries worldwide, saw a significant shift in public health strategies with the emergence of the COVID-19 pandemic. Hand hygiene became one of the several measures introduced during the pandemic in Ghana by the government. Scientific evidence indicates that hand hygiene is one of the most efficient measures of controlling infections [30,31]. For instance, a Cochrane meta-analysis brought to light sufficient evidence that hand hygiene is a very cost-effective preventive measure in reducing gastrointestinal infections including diarrheal diseases [32,33].

This study examined the relationship between the incidence of diarrhea cases in Ghana and the adoption of hand hygiene throughout the COVID-19 period. The data spanned from February 2018 to March 2022, incorporating the pre- hand hygiene protocol and hand hygiene protocol periods to discern trends in diarrhea cases nationwide and across various regions within Ghana.

The findings from both descriptive statistics from national and sub-national analyses revealed a notable reduction in diarrheal cases following the COVID-19 hand hygiene regime.

The study's findings demonstrate a significant decrease in diarrheal cases following the institution of COVID-19 hand hygiene protocols in Ghana. Twelve (12) out of the 16 regions saw significant reductions of diarrhea cases with the advent of the protocols. This agrees with a study conducted in Ethiopia with a nationwide reduction in the prevalence of diarrhea and pneumonia cases among under five children and variation in the administrative regions after hand hygiene practice was instituted [34].

The national-level study revealed a statistically significant difference between the average number of reported cases of diarrhea before and during the COVID-19 handwashing regimen's adoption. The observed reduction of 13,463 cases (95% CI [5847.19, 21,078.81]) during the protocol implementation period reflects a substantial effect, suggesting a potential positive influence of enhanced hygiene practices on reducing diarrhoea morbidity [35].

Sub-national analyses further revealed regional variations on the effect of the hand hygiene protocol on diarrhea cases. Greater Accra region had biggest percentage reduction with 24% of the total diarrhea cases. This could be because, Greater Accra region was the hardest-hit region by COVID-19 [7,36], as compared to the other regions, hence it had more access to hand hygiene facilities and associated sensitization than the other regions or been more precautious than the others due to the number of COVID-19 cases and deaths occurring around them. During the COVID-19 pandemic, the Greater Accra was termed as the "hotspot" area which led to the implementation of most of the COVID-19 restrictions such as lockdowns, restriction on public gatherings, the closure of borders in these regions for public safety and the institution of mandatory hand hygiene protocols at all public places [36,37]. Ghanaians residing in the Greater Accra region exhibited an increased level of consciousness regarding personal hygiene, while a subset of individuals held skepticism towards the legitimacy of these efforts [38]. Ghanaians had never been the best at washing their hands, even before the

**Table 3. Causal impact of the introduction of COVID-19 hand hygiene protocol on diarrhea cases in Ghana.**

| Region | Average | Prediction (95% CI) | Absolute Effect (95% CI) | Cumulative | Prediction (95% CI) | Absolute Effect (95% CI) | Relative Effect (95% CI) | P-value | Prob. of Causal Effect |
|---|---|---|---|---|---|---|---|---|---|
| Ahafo | 3827 | 4095 (3930, 4252) | -268 (-425, 102) | 95682 | 102380 (98243, 106295) | -6698 (-10613, -2561) | -6.5% (-10%, -2.6%) | **0.001** | **99.9%** |
| Ashanti | 13357 | 15268 (13302, 17138) | -1910 (-3781, 55) | 333937 | 381691 (332557, 428451) | -47754 (-94514, 1380) | -12% (-22%, 0.4%) | **0.031** | **96.9%** |
| Bono | 7276 | 7706 (7313, 8099) | -430 ( -823, -37) | 181898 | 192652 (182826, 202475) | -10754 (-20577, -928) | -5.6% (-10%, -0.5%) | **0.020** | **98.0%** |
| Bono East | 7444 | 7408 (6959, 7855) | 36 (-411, 485) | 186100 | 185191 (173978, 196385) | 909 (-10285, 12122) | 0.5% (-5.2%, 7.0%) | 0.421 | 58.0% |
| Central | 7759 | 8644 (8189, 9081) | -885 (-1322, -430) | 193978 | 216097 (204729, 227026) | -22119 (-33048, -10751) | -10% (-15%, -5.3%) | **0.001** | **99.9%** |
| Eastern | 12520 | 14133 (13497, 14769) | -1614 (-2249, -977) | 312995 | 353333 (337423, 369222) | -40338 (-56227, -24428) | -11.0% (-15%, -7.2%) | **0.001** | **99.9%** |
| Greater Accra | 7170 | 9409 (8782, 10089) | -2239 (-2919, -1612) | 179248 | 235219 (219555, 252234) | -55971 (-72986, -40307) | -24% (-29%, -18%) | **0.001** | **99.9%** |
| North East | 4000 | 3852 (3599, 4090) | 196 (-42, 449) | 100000 | 96293 (89976, 102253) | 4907 (-1053, 11224) | 5.3% (-1.0%, 12.0%) | 0.058 | 94.0% |
| Northern | 7426 | 9194 (8759, 9661) | -1768 (-2235, -1333) | 185654 | 229849 (218968, 241519) | -44195 (-55865, -33314) | -19% (-23%, -15%) | **0.001** | **99.9%** |
| Oti | 3233 | 3365 (3168, 3559) | -131 (-326, 65) | 80831 | 84115 (79210, 88970) | -3284 (-8139, 1621) | -3.8% (-9.1%, 2.0%) | 0.088 | 91.0% |
| Savannah | 2550 | 2670 (2531, 2809) | -120 (-259, 19) | 63748 | 66746 (63274, 70223) | -2998 (-6475, 474) | -4.4% (-9.2%, 0.8%) | **0.044** | **95.6%** |
| Upper East | 5984 | 6738 (6251, 7272) | -754 (-1288, -267) | 149593 | 168453 (156268, 181799) | -18860 (-32206, -6675) | -11.0% (-18%, -4.3%) | **0.003** | **99.7%** |
| Upper West | 4876 | 5414 (5056, 5760) | -537 (-884, -179) | 121912 | 135345 (126392, 144000) | -13433 (-22088, -4480) | -9.8 (-15%, -3.5%) | **0.003** | **99.7%** |
| Volta | 4727 | 5514 (5122, 5900) | -787 (-1174, -395) | 118164 | 137851 (128045, 147504) | -19687 (-29340, -9881) | -14.0% (-20%, -7.7%) | **0.001** | **99.9%** |
| Western | 9271 | 9935 (9591, 10274) | -665 (-1004, -321) | 231765 | 248387 (239787, 256861) | -16622 (-25096, -8022) | -6.6% (-9.8%, -3.3%) | **0.001** | **99.9%** |
| Western North | 4711 | 5448 (4292, 6689) | -737 (-1978, 419) | 117774 | 136203 (107290, 167225) | -18429 (-49451, 10484) | -13.0% (-30%, 9.8%) | 0.112 | 89.0% |
| **Ghana** | 106179 | 119283 (114997, 123482) | -13103 (-17303, -8817) | 2654479 | 2982065 (2874915, 3087054) | -327586 (-432575, -220436) | -11.0% (-14%, -7.7%) | **0.001** | **99.9%** |

COVID-19 pandemic started. For example, some studies on the habits of Ghanaian adolescents in Accra found that most young people did not follow a hand hygiene practice [39,40].

According to Kyei-Arthur et al. (2023) regular washing of hands was the easiest COVID-19 preventive protocol practiced by individuals. This encouraged high compliance with the hand hygiene protocol. In Ghana, audio communications were used to address personal hygiene and hand washing in transport terminals, markets and other public areas within the regions

[5]. Even though the number of hospital admissions for non-COVID-19 patients decreased, residents of communities showed initiative in improving their health. This strict adherence to established guidelines and a heightened emphasis on self-care practices may have resulted in the most substantial reduction in diarrheal diseases in these regions.

Several regions, including Ahafo, Central, Eastern, Greater Accra, Northern, Upper East, Upper West, and Volta, recorded statistically significant reductions in diarrheal cases during the protocol period. These reductions ranged from 6.5% to 24%, signifying the effectiveness of hand hygiene interventions in reducing diarrheal diseases within these regions.

It is crucial to remember, nevertheless, that not all regions saw statistically significant decreases. Some regions, such as Bono East, North-East, Oti, and Western North, demonstrated non-significant changes in diarrheal cases. This may be because these are newly created regions from the already existing 10 regions in Ghana and therefore the results observed might stem from variations in socio-economic factors, healthcare infrastructure, cultural practices, or other unmeasured contextual variables affecting the efficacy of the hand hygiene protocol in these areas.

By highlighting the importance of the handwashing regimen in preventing an estimated 11.0% (95% CI: -14%, -7.6%) of diarrhea cases worldwide, the causal effect analysis validated the reported reductions in diarrheal cases. The varied regional effects underscore the nuanced nature of the intervention's effectiveness across different parts of Ghana.

The strengths of this paper are found in its detailed analysis that covers both national and sub-national levels and offers a nuanced understanding of the effect of the intervention. The absence of a control group and the use of OPD attendance as the possible confounding variable are some of limitations that could have an impact on the reported declines because of the use of secondary data.

## Conclusion

The findings suggest an association between the implementation of COVID-19 hand hygiene protocols and reduced diarrheal disease burden in Ghana. This study demonstrates that the hand hygiene regimen that was implemented during COVID-19 in Ghana led to an 11% decrease in diarrheal infections, with regional variations. Twelve (12) out of the 16 regions showed significant reductions in diarrhea cases with the Greater Accra region (the hotspot of COVID-19 in Ghana) recording the highest reduction in diarrheal diseases during this period. Sustained and targeted public health interventions on hand hygiene practices are advocated to further mitigate diarrheal morbidity, particularly in regions where the effect was less pronounced.

## Supporting information

**S1 Data.**
(XLS)

**S2 Data.**
(LOG)

## Author Contributions

**Conceptualization:** George Asumah Adu, Henry Ofosu Addo, Nana Yaw Peprah, Ernest Kenu, Franklin Asiedu Bekoe, Keziah L. Malm.

**Data curation:** Kingsley E. Amegah, Henry Ofosu Addo, Theresa Andoh, Ferguson Duvor, Gladys Antwi.

**Formal analysis:** George Asumah Adu, Kingsley E. Amegah, Theresa Andoh.

**Investigation:** George Asumah Adu, Theresa Andoh, Ferguson Duvor, Keziah L. Malm.

**Methodology:** George Asumah Adu, Kingsley E. Amegah, Henry Ofosu Addo, Theresa Andoh, Gladys Antwi.

**Project administration:** Franklin Asiedu Bekoe.

**Software:** George Asumah Adu, Kingsley E. Amegah.

**Supervision:** George Asumah Adu, Nana Yaw Peprah, Ernest Kenu, Franklin Asiedu Bekoe, Keziah L. Malm.

**Validation:** Henry Ofosu Addo, Ernest Kenu.

**Visualization:** George Asumah Adu, Kingsley E. Amegah, Ferguson Duvor, Gladys Antwi, Keziah L. Malm.

**Writing – original draft:** George Asumah Adu, Kingsley E. Amegah, Henry Ofosu Addo, Theresa Andoh, Ferguson Duvor, Gladys Antwi, Nana Yaw Peprah.

**Writing – review & editing:** George Asumah Adu, Kingsley E. Amegah, Henry Ofosu Addo, Theresa Andoh, Ferguson Duvor, Gladys Antwi, Nana Yaw Peprah, Ernest Kenu, Franklin Asiedu Bekoe, Keziah L. Malm.

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
