## [Decision Letter · Decision Letter 0]

18 Jun 2024

PONE-D-24-12201Killing “two birds” with one stone: Impact of COVID-19 hand hygiene protocols on diarrheal cases in GhanaPLOS ONE

Dear Dr. Adu,

Thank you for submitting your manuscript to PLOS ONE. After careful consideration, we feel that it has merit but does not fully meet PLOS ONE’s publication criteria as it currently stands. Therefore, we invite you to submit a revised version of the manuscript that addresses the points raised during the review process. <please by="" manuscript="" revised="" submit="" your="">Please submit your revised manuscript by Aug 02 2024 11:59PM. If you will need more time than this to complete your revisions, please reply to this message or contact the journal office at plosone@plos.org. Please include the following items when submitting your revised manuscript:</please>A rebuttal letter that responds to each point raised by the academic editor and reviewer(s). You should upload this letter as a separate file labeled 'Response to Reviewers'.A marked-up copy of your manuscript that highlights changes made to the original version. You should upload this as a separate file labeled 'Revised Manuscript with Track Changes'.An unmarked version of your revised paper without tracked changes. You should upload this as a separate file labeled 'Manuscript'.If applicable, we recommend that you deposit your laboratory protocols in protocols.io to enhance the reproducibility of your results. Protocols.io assigns your protocol its own identifier (DOI) so that it can be cited independently in the future. For instructions see: https://journals.plos.org/plosone/s/submission-guidelines#loc-laboratory-protocols. Additionally, PLOS ONE offers an option for publishing peer-reviewed Lab Protocol articles, which describe protocols hosted on protocols.io. Read more information on sharing protocols at https://plos.org/protocols?utm_medium=editorial-email&utm_source=authorletters&utm_campaign=protocols.

We look forward to receiving your revised manuscript.

Kind regards,

Khin Thet Wai, MBBS, MPH, MA

Academic Editor

PLOS ONE

Journal Requirements:

2. You indicated that ethical approval was not necessary for your study. We understand that the framework for ethical oversight requirements for studies of this type may differ depending on the setting and we would appreciate some further clarification regarding your research. Could you please provide further details on why your study is exempt from the need for approval and confirmation from your institutional review board or research ethics committee (e.g., in the form of a letter or email correspondence) that ethics review was not necessary for this study? Please include a copy of the correspondence as an ""Other"" file.

3. In the online submission form, you indicated that The data underlying the results presented in the study are available from the corresponding author upon reasonable request.

4. We note that Figure 1 in your submission contain map images which may be copyrighted. All PLOS content is published under the Creative Commons Attribution License (CC BY 4.0), which means that the manuscript, images, and Supporting Information files will be freely available online, and any third party is permitted to access, download, copy, distribute, and use these materials in any way, even commercially, with proper attribution. For these reasons, we cannot publish previously copyrighted maps or satellite images created using proprietary data, such as Google software (Google Maps, Street View, and Earth). For more information, see our copyright guidelines: http://journals.plos.org/plosone/s/licenses-and-copyright.

We require you to either present written permission from the copyright holder to publish these figures specifically under the CC BY 4.0 license, or remove the figures from your submission:

Reviewers' comments:

Reviewer's Responses to Questions

**Comments to the Author**

1. Is the manuscript technically sound, and do the data support the conclusions?

Reviewer #1: Partly

Reviewer #2: Yes

2. Has the statistical analysis been performed appropriately and rigorously? 

Reviewer #1: I Don't Know

Reviewer #2: Yes

3. Have the authors made all data underlying the findings in their manuscript fully available?

Reviewer #1: No

Reviewer #2: Yes

4. Is the manuscript presented in an intelligible fashion and written in standard English?

Reviewer #1: Yes

Reviewer #2: Yes

5. Review Comments to the Author

**Reviewer #1:** GENERAL REMARKS

The paper pinpoints the contribution of hand washing introduced during the COVID-19 to a decrease in reported diarrhea cases in Ghana. The paper is generally well written and flows very well.

The key concern about the paper is the use of the word “impact” as this suggests causality. The authors use the word impact based on the statistical package used. Even though the statistical package may have this property, I’m not sure there is guarantee that all factors that could have contributed to the reduction in diarrhea cases were accounted for. This is especially considering that that in the sub-national analyses, not all the regions showed a significant reduction in diarrhea cases.

It is suggested that “association” is used instead of the word “impact” in the paper.

The authors also need to provide references for some statements made and these areas are highlighted in the paper.

TITLE

For reasons given above it is suggested that “association” is used instead of the word “impact” which suggests causality but in my view is not proven beyond doubt.

INTRODUCTION

The introduction is well written but the same comment applies to the use of the word “impact on page 3 Line 89.

METHODS

Lines 140 on page 5. Please provide more information on the incorporated covariates

Statistical Analysis

In reference to Lines 133 to 138 on page 5.

The likelihood is that some readers may not fully understand this narrative (especially the term counterfactual scenario) or may not be familiar with the Causal Impact package.

I suggest that this section on the statistical analyses is simplified to enable readers really appreciate the point that it was indeed the introduction of the hand washing that caused the decrease in diarrhea cases. Better still I propose association instead of impact because there is no guarantee all factors were accounted for.

Lines 141-142 Please provide information on the OPD data and explain how the OPD attendance was controlled for.

RESULTS

Lines 149, 156 Please provide references

Lines 185 to 204 Please present data in a table

DISCUSSION

1.In reference to these statements:

It is crucial to remember, nevertheless, that not all locations saw statistically significant

decreases. Some regions, such as Bono East, North-East, Oti, and Western North, demonstrated

non-significant changes in diarrheal cases.

.

On page 19 in the Discussion, an argument was made that Greater Accra and Ashanti showed higher reductions in diarrhea cases and were the hardest hit.

Following that argument please comment on whether these regions that didn't show significant reductions were the least hit.

2.In reference to the following statement:

These disparities might stem from variations in socio-economic factors, healthcare infrastructure, cultural practices, or other unmeasured contextual variables impacting the efficacy of the hand hygiene protocol in these areas.

What is the evidence to suggest this? Please back it up with evidence from literature and references.

CONCLUSION

In line with this first statement of the conclusion

The findings suggest an association between the implementation of COVID-19 handwashing

protocols and reduced diarrheal disease burden in Ghana.

The suggestion is to use "associated" instead of "impact" in the paper since causality (hand washing causing a degree in diarrhea cases) cannot be 100% proven as all possible factors were not controlled for.

**Reviewer #2:** General Comment

Overall, the topic is interesting and the manuscript is well-written. The main results highlight the reduction in diarrheal cases in Ghana due to the COVID-19 hand hygiene protocol, using observed data from the pre-protocol and protocol periods as well as a counterfactual model in the "causal impact R package (Bayesian Structural Time Series Model)." Please check the entire manuscript for consistency and to avoid typos.

Specific Comments and Suggestions

1.In the “Statistical Analysis” section under “METHODS,” the author mentioned that a two-sample t-test was conducted. Were any other statistical tests, in addition to the two-sample t-test, applied? The Mann-Whitney test was observed in some figures comparing summary measures between the pre-protocol and protocol periods (Figure 2-A to D).

2.The author mentioned that OPD attendance was considered a potential confounder and controlled for (line number 141), but in the “DISCUSSION” section, the fact that potential confounding variables were not considered throughout the study is reported as one of the limitations.

3.In Table 1 of the “RESULTS” section, please recheck the numbers in the “Sum” column, as the addition of diarrheal cases during the pre-protocol and protocol periods slightly differs from the total cases in Ghana.

4.In line numbers 176 and 207, the figure numbers referenced by the author are incorrect (Figure 1 was incorrectly mentioned instead of Figure 2, as Figure 1 refers to the map of Ghana).

5.When presenting the results for the “Sub-national analysis of the difference in diarrhea cases prior to and during the introduction of the COVID-19 handwashing protocol in Ghana,” I suggest presenting these results in a table (similar to Table 2 for presenting causal impact) for better readability and clarity. A table will not duplicate the figures (Figure 2, 2A to 2D) because it will include the mean difference with 95% CI, t-statistics, p-value, and Cohen’s d with 95% CI, which are not presented in the current figures.

6.I suggest conducting a subgroup analysis by age strata if data is available. (In DHIMS 2, if the age-group variable of the aggregate data for diarrheal cases is available, the results for the reduction in diarrheal cases for specific age groups, such as under-five, would be more interesting.)

6. PLOS authors have the option to publish the peer review history of their article (what does this mean?). If published, this will include your full peer review and any attached files.

Reviewer #1: No

Reviewer #2: No

---

## [Author Response · Author response to Decision Letter 0]

23 Jul 2024

Response to Reviewers letter has been uploaded.

---

## [Decision Letter · Decision Letter 1]

8 Aug 2024

Reduction in Diarrhea cases following Implementation of COVID-19 hand hygiene Interventions in Ghana: A Causal Impact Analysis

PONE-D-24-12201R1

Dear Dr. Adu,

We’re pleased to inform you that your manuscript has been judged scientifically suitable for publication and will be formally accepted for publication once it meets all outstanding technical requirements.

Kind regards,

Khin Thet Wai, MBBS, MPH, MA

Academic Editor

PLOS ONE

Additional Editor Comments (optional):

All comments are adequately addressed.

Reviewers' comments:

Reviewer's Responses to Questions

**Comments to the Author**

1. If the authors have adequately addressed your comments raised in a previous round of review and you feel that this manuscript is now acceptable for publication, you may indicate that here to bypass the “Comments to the Author” section, enter your conflict of interest statement in the “Confidential to Editor” section, and submit your "Accept" recommendation.

Reviewer #2: All comments have been addressed

2. Is the manuscript technically sound, and do the data support the conclusions?

Reviewer #2: Yes

3. Has the statistical analysis been performed appropriately and rigorously? 

Reviewer #2: Yes

4. Have the authors made all data underlying the findings in their manuscript fully available?

Reviewer #2: Yes

5. Is the manuscript presented in an intelligible fashion and written in standard English?

Reviewer #2: Yes

6. Review Comments to the Author

Reviewer #2: I would like to appreciate the authors for responding to most of my comments from the first round with precise action.

Please check to avoid the grammatical errors; at line number 202 to 204, the authors mentioned statistically significant reductions were found in most of the regions but the mean difference for only “Central” region was described; “Statistically significant reductions (Mean difference = 1224.4, CI: 631.5 – 1817.3, p = 0.001, Cohen’s d = 1.17) in diarrhea cases were observed in the Central, Eastern, Greater Accra, Northern, Upper East, Upper West, Volta, and Western regions, with varying effect sizes,……..”. Moreover, please check to ensure the consistency; the discrepancy between the table and text was observed; for an instance; at line number 211, p-value was written as “0.000” but in the table, the respective p-value was “0.001”.

7. PLOS authors have the option to publish the peer review history of their article (what does this mean?). If published, this will include your full peer review and any attached files.

Reviewer #2: No

---

## [Editor Report · Acceptance letter]

21 Aug 2024

PONE-D-24-12201R1 

PLOS ONE

Dear Dr. Adu, 

I'm pleased to inform you that your manuscript has been deemed suitable for publication in PLOS ONE. Congratulations! Your manuscript is now being handed over to our production team.

Kind regards, 

on behalf of

Dr. Khin Thet Wai 

Academic Editor

PLOS ONE